# Learning Junta Distributions, Quantum Junta States, and QAC$^0$ Circuits

**Jinge Bao** [1] [2]   **Francisco Escudero Gutiérrez** [3]

## Abstract

In this work, we consider the problems of learning junta distributions, their quantum counterparts (quantum junta states), and QAC$^0$ circuits, which we show to be close to juntas. (i) **Junta distributions.** A probability distribution $p : \{-1,1\}^n \rightarrow [0,1]$ is a $k$-junta if it only depends on $k$ bits. We show that they can be learned to within additive error $\varepsilon$ in total variation distance by $O(2^k \log(n)/\varepsilon^2)$ samples, which quadratically improves the upper bound given by Aliakbarpour, Blais, and Rubinfeld (COLT'16) and matches their lower bound in every parameter. (ii) **Junta states.** We initiate the study of $n$-qubit states that are $k$-juntas, those that are the tensor product of a $k$-qubit state and an $(n-k)$-qubit maximally mixed state. We show that these states can be learned with error $\varepsilon$ in trace distance with $O(12^k \log(n)/\varepsilon^2)$ single copies. We also prove a lower bound of $\Omega((4^k + \log(n))/\varepsilon^2)$ copies. Additionally, we show that, for constant $k$, $\tilde{\Theta}(2^n/\varepsilon^2)$ copies are necessary and sufficient to test whether a state is $\varepsilon$-close or $7\varepsilon$-far from being a $k$-junta. (iii) QAC$^0$ **circuits.** Nadimpalli et al. (STOC'24) recently showed that the Pauli spectrum of QAC$^0$ circuits (with a limited number of auxiliary qubits) is concentrated on low degrees. We remark that they implied something stronger, namely that the Choi states of those circuits are close to being juntas. As a consequence, we show that $n$-qubit QAC$^0$ circuits with size $s$, depth $d$ and $a$ auxiliary qubits can be learned from $2^{O(\log(s^2 2^a)^d)} \log(n)$ copies of the Choi state, improving the $n^{O(\log(s^2 2^a)^d)}$ by Nadimpalli et al. Along the way, we give a new proof of the optimal performance of Classical Shadows based on Pauli

analysis. We also strengthen the lower bounds against QAC$^0$ to compute the address function. Our techniques are based on Fourier and Pauli analysis, and our learning upper bounds are a refinement of the low-degree algorithm by Linial, Mansour, and Nisan.

## 1. Introduction

One of the main questions of computational learning theory is how efficiently we can learn an unknown object that is promised to have some structure. Two of the most studied structured objects are juntas, which are multi-bit or multi-qubit objects where only a few of the bits or qubits are relevant, and constant-depth circuits. There is plenty of literature about learning junta objects, such as junta Boolean functions (Atıcı & Servedio, 2007; Ambainis et al., 2016), junta distributions (Aliakbarpour et al., 2016; Chen et al., 2021), quantum junta unitaries (Chen et al., 2023), and quantum junta channels (Bao & Yao, 2025). Two celebrated models of constant-depth circuits that have been studied from the point of view of learning are AC$^0$ circuits (Linial et al., 1993; Eskenazis et al., 2023) and their quantum analogue, QAC$^0$ circuits (Nadimpalli et al., 2024; Anshu et al., 2025; Vasconcelos & Huang, 2025).

We continue this line of research by improving the upper bounds on learning classical junta distributions and QAC$^0$ circuits. To bridge junta distributions and QAC$^0$ circuits, we initiate the first study on quantum junta states. All of our upper bounds exploit the observation that the Fourier/Pauli expansions of the considered objects are close to being supported on a few low-degree characters/Pauli operators. In other words, these objects are both low-degree and sparse. In the case of juntas, both properties follow from the definition. In the case of QAC$^0$ circuits, they were shown to concentrate on low degrees in (Nadimpalli et al., 2024). In this work, we show that they are also close to juntas.

### 1.1. Our results

Our learning upper bounds are summarized in Table 1, where $n$ is the number of bits or qubits, $k$ stands for the number of relevant variables of junta objects, $s$ (which stands for size) is the number of multi-qubit gates of a QAC$^0$ cir-

[1]School of Informatics, University of Edinburgh, Edinburgh, United Kingdom [2]Centre for Quantum Technologies, Singapore [3]Qusoft and CWI, Amsterdam, the Netherlands. Correspondence to: Jinge Bao <jinge.bao@ed.ac.uk>, Francisco Escudero Gutiérrez <feg@cwi.nl>.

*Proceedings of the 43$^{rd}$ International Conference on Machine Learning*, Seoul, South Korea. PMLR 306, 2026. Copyright 2026 by the author(s).

cuit, $d$ is the depth of a circuit, and $\varepsilon$ is the error parameter with respect to natural metrics that we will specify later. In the case of classical objects, the complexity measure is the number of samples. In the quantum case, the complexity measure is the number of copies.

Before we discuss our results in more detail, we make a few remarks about our main results.

(i) **Junta distributions.** Our upper bound is essentially optimal, as it matches the lower bound $\Omega(2^k/\varepsilon^2 + k\log(n)/\varepsilon)$ of (Aliakbarpour et al., 2016) in every parameter (see Theorem 1.1).

(ii) **Junta states.** Recent works study the junta-learning problem of unitaries and quantum channels (Chen et al., 2023; Bao & Yao, 2025), but there seems to be no previous work about quantum junta states. Hence, our result for junta states fills a gap in the literature. We also provide a $\Omega((4^k + \log(n))/\varepsilon^2)$ lower bound that shows that our upper bound cannot be improved by much (see Theorem 1.3).

(iii) $\mathsf{QAC}^0$ **circuits.** For constant $d$, $s$ and $\varepsilon$, our result exponentially improves previous work (see Theorem 1.5). However, in the usual regime where $s = \mathrm{poly}(n)$ and $d, \varepsilon$ are constants, it yields a quasi-polynomial number of samples, which was already attained in previous work (Nadimpalli et al., 2024).

Additionally, we show that, for constant $k$, $\tilde{\Theta}(2^n/\varepsilon^2)$ copies are necessary and sufficient to test whether a state is $\varepsilon$-close or $7\varepsilon$-far from being a $k$-junta.

### 1.1.1. LEARNING JUNTA DISTRIBUTIONS

Learning in the presence of irrelevant information, such as the *dummy* variables appearing in juntas, is one of the most famous yet open problems of classical computational learning theory since the 90's (Blum, 1994; Blum et al., 1995; Blum & Langley, 1997). Numerous works consider the problems of learning and testing junta Boolean functions and distributions (Mossel et al., 2003; Valiant, 2012; Aliakbarpour et al., 2016; Chen et al., 2021; 2024; Nadimpalli & Patel, 2024). In particular, Aliakbarpour, Blais, and Rubinfeld considered the problem of learning a junta distribution $p\colon \{-1,1\}^n \to [0,1]$ from given samples $x \sim p(x)$ (Aliakbarpour et al., 2016). They showed that in order to estimate $p$ to within additive error $\varepsilon$ in total variation distance, $O(2^{2k}k\log(n)/\varepsilon^4)$ samples suffice and $\Omega(2^k/\varepsilon^2 + k\log(n)/\varepsilon)$ are necessary. As the first result of this work, we quadratically improve the learning upper bound compared to theirs. Moreover, it matches their lower bound in every parameter.

**Theorem 1.1.** *Let $p\colon \{-1,1\}^n \to [0,1]$ be a $k$-junta distribution. The distribution can be learned to within additive*

*error $\varepsilon$ in total variation distance with success probability at least $1 - \delta$ by $O(\frac{2^k k \log(n/\delta)}{\varepsilon^2})$ samples from $p$.*

Similar to the technique for proving the upper bound by Aliakbarpour et al, we leverage the fact that $k$-juntas have Fourier degree at most $k$. Furthermore, we observe that the Fourier spectrum is also sparse. Our algorithm is proper (i.e., it outputs a probability distribution) and its time complexity is $O(n^k 2^k/\varepsilon^2)$ (see Remark 2.1), which also quadratically improves the upper bound of time complexity $O(n^k 2^{2k}/\varepsilon^4)$ by Aliakbarpour et al.

### 1.1.2. TESTING AND LEARNING JUNTA STATES

In quantum testing and learning theory, the most commonly studied objects are states, unitaries, and channels (O'Donnell & Wright, 2021; Haah et al., 2017; 2023; Oufkir, 2023). By contrast, the problems of learning and testing $k$-junta unitaries and channels were recently studied (Chen et al., 2021; Bao & Yao, 2025), but to the best of our knowledge, no literature has explored the analogue version for quantum states.[1]

**Definition 1.2** (Junta state). A $n$-qubit state $\rho$ is said to be a $k$-junta state if there are a set $K \subseteq [n]$ of size $k$ and a state $\rho_K$ defined on $K$ such that

$$\rho = \rho_K \otimes \frac{I_{[n]-K}}{2^{n-k}}.$$

In other words, $\rho$ is a $k$-junta state if $\rho$ is the tensor product of a nontrivial $k$-qubit state and the maximally mixed state on the rest $(n - k)$ qubits.

Note that $k$-junta states are the quantum generalizations of $k$-junta distributions. Therefore, the problems of learning and testing quantum junta states are the quantum analogue of the problems considered by Aliakbarpour et al. (Aliakbarpour et al., 2016). We prove nearly optimal results for testing and learning quantum junta states in terms of copy complexity.

**Theorem 1.3.** *Let $\rho$ be a $n$-qubit $k$-junta quantum state. Then, $\rho$ can be learned to within additive error $\varepsilon$ in trace distance and success probability at least $1 - \delta$, by $O(\frac{12^k \log(n/\delta)}{\varepsilon^2})$ copies of $\rho$, and $\Omega(\frac{\log(n)+4^k}{\varepsilon^2})$ are necessary for this task. Furthermore, the algorithm just does Pauli measurements on single copies of the state.*

For the upper bound, we perform classical shadow tomography with Pauli measurements (Huang et al., 2020; Elben et al., 2022). Furthermore, we include a novel proof of the rigorous guarantees of the classical shadow tomography algorithm based on Pauli analysis that might be of independent interest (see Theorem 3.1). The second term of

---

[1]An incomparable notion of $k$-junta states was explored in (Zhao et al., 2024). Those states are pure states where all but $k$ registers equal $|0\rangle$.

*Table 1.* Summary of learning bounds.

| TYPE | PREVIOUS RESULT | OUR RESULT |
|---|---|---|
| JUNTA DISTRIBUTIONS | $O(2^{2k}\log(n)/\epsilon^4)$ (ALIAKBARPOUR ET AL., 2016) $\Omega(2^k/\epsilon^2 + k\log(n)/\epsilon)$ (ALIAKBARPOUR ET AL., 2016) | $O(2^k\log(n)/\epsilon^2)$ THEOREM 1.1 |
| QUANTUM JUNTA STATES | — | $O(12^k\log(n)/\epsilon^2)$ THEOREM 1.3 |
| | — | $\Omega((4^k + \log(n))/\epsilon^2)$ THEOREM 1.3 |
| QAC$^0$ CIRCUITS | $O(n^{\log(s/\epsilon)^d})$ (NADIMPALLI ET AL., 2024) | $O(2^{\log(s/\epsilon)^d}\log(n))$ THEOREM 1.5 |

the proved lower bound $\Omega(4^k/\varepsilon^2)$ follows from the lower bound by Haah et al. to learn $k$-qubit states (Haah et al., 2017). In order to prove the first term of the lower bound $\Omega(\log(n)/\varepsilon^2)$, we show that there are $n$ states which are 1-junta but hard to distinguish.

For the testing problem, we have the following result,

**Theorem 1.4.** *Let $\rho$ be a $n$-qubit state. Let $k \in \mathbb{N}$ be a constant. Then $\widetilde{\Theta}\big(\frac{2^n \log(1/\delta)}{\varepsilon^2}\big)$ copies of $\rho$ are necessary and sufficient to test whether $\rho$ is $\varepsilon$-close or $7\varepsilon$-far in trace distance from being $k$-junta with success probability at least $1 - \delta$.*[2]

For the upper bound, we use the quantum state certification algorithm of Bădescu, O'Donnell, and Wright (Bădescu et al., 2019). For the lower bound, we reduce the testing junta-ness to testing whether a state is maximally mixed (O'Donnell & Wright, 2021).

### 1.1.3. LEARNING QAC$^0$ CIRCUITS

The QAC$^0$ circuits were proposed by Moore as the quantum analogue of AC$^0$ circuits (Moore, 1999). In that work, Moore asked whether QAC$^0$ circuits can compute parity. Despite various efforts, the question remains open to date (Fang et al., 2006; Rosenthal, 2021; Nadimpalli et al., 2024; Anshu et al., 2025; Fenner et al., 2025). In a recent work, Nadimpalli, Parham, Vasconcelos, and Yuen made progress by showing that the Pauli spectrum of the Choi state of a QAC$^0$ circuit with not too many auxiliary qubits satisfied low-degree concentration (Nadimpalli et al., 2024). In addition, we note that the Choi state of a QAC$^0$ circuit is not only concentrated on low-degree operators, but also close to being a quantum junta state (see Theorem 4.1). Based on this new observation, alongside the algorithm of Theorem 1.3, we can prove a stronger result.

**Theorem 1.5.** *Let $\rho$ be the Choi state of a $n$-qubit QAC$^0$ circuit with size $s$, depth $d$, and $a$ auxiliary qubits. Then us-*

ing $2^{O((\log(s^2 2^a/\varepsilon))^d)} \log(n/\delta)$ *copies of $\rho$, one can output a $\rho'$ such that with probability at least $1 - \delta$ satisfies*

$$2^n \|\rho - \rho'\|_F^2 \leq \varepsilon.$$

*Furthermore, the algorithm just does Pauli measurements on single copies of the state.*

The only previous result on learning QAC$^0$ circuits was Theorem 39 in (Nadimpalli et al., 2024), and our Theorem 1.5 improves it from $n^{O((\log(s^2 2^a/\varepsilon))^d)}$ copies to $2^{O((\log(s^2 2^a/\varepsilon))^d)} \log(n)$ copies.[3] At this point, it might be unclear why we have chosen to learn the Choi state of the circuit in the $2^n$-Frobenius norm. This is the same figure of merit as the one considered in (Nadimpalli et al., 2024) (the authors of that work use a slightly different notation), and in Section 4.1 we explain the reason why it is natural to consider it for this learning task.

In addition, in Section 4.4 we use that QAC$^0$ are close to juntas, and not only to low-degree, to show new lower bounds for computing the address function, which is the canonical example of a low-degree function that depends on many variables.

**Paper organization**. In the rest of the main text, we include the details of the stated results, and we defer the preliminaries and some proofs to the appendix. In the appendix, we also include a section that we title *Our algorithms in a nutshell,* where we provide an intuition of why our algorithms improve over previous works that only use low-degree concentration and not sparsity.

---

[2]Here, $\widetilde{\Theta}(\cdot)$ hides poly-log factors in the argument.

[3]In a concurrent work, Huang and Vasconcelos extend this result to recover not only the Choi-state but also the unitary defined by the circuit (Vasconcelos & Huang, 2025). We also note that even for constant $s$ and constant $a$, the number of 1-qubit gates of the circuit can be of the order $nd$, so the results for learning circuits of bounded gate complexity of (Zhao et al., 2024) yield upper bounds of the kind $O(n)$, worse than ours $O(\log(n))$ for this regime.

## 2. Learning Junta Distributions

In this section, we prove Theorem 1.1, which we restate for the reader's convenience.

**Theorem 1.1.** *Let $p\colon \{-1,1\}^n \to [0,1]$ be a k-junta distribution. The distribution can be learned to within additive error $\varepsilon$ in total variation distance with success probability at least $1 - \delta$ by $O(\frac{2^k k \log(n/\delta)}{\varepsilon^2})$ samples from $p$.*

We begin by recalling what the usual model for learning distributions is. Given a distribution $p\colon \{-1,1\}^n \to [0,1]$, one can access it by sampling $x \in \{-1,1\}^n$ with probability $p(x)$. The goal of the learner is to use a few samples to output another distribution $p'\colon \{-1,1\}^n \to [0,1]$ that is $\varepsilon$-close to $p$ in total variation distance, which is given by

$$d_{\mathrm{TV}}(p,p') = \frac{1}{2}\|p - p'\|_{\ell_1} = \frac{1}{2} \sum_{x \in \{-1,1\}^n} |p(x) - p'(x)|.$$

If $p\colon \{-1,1\}^n \to [0,1]$ is a $k$-junta depending on the variables of a set $K \subseteq [n]$ of size $k$, then it can be written as

$$p(x) = \sum_{S \subseteq K} \widehat{p}(S) \prod_{i \in S} x_i,$$

where $\widehat{p}(S) = \mathbb{E}_{x \in \{-1,1\}^n} p(x) \prod_{i \in S} x_i$ are the Fourier coefficients of $p$. Note that all non-zero Fourier coefficients of a $k$-junta correspond to monomials of degree $\leq k$, and there are at most $2^k$ of them. We use this to show a nearly optimal algorithm to learn $k$-junta distributions.

*of Theorem 1.1.* Let $T = O(\frac{2^k}{\varepsilon^2} k \log(\frac{n}{\delta}))$ be the number of samples $(x^1, \ldots, x^T)$ we take. For every $S \subseteq [n]$ with $|S| \leq k$ we define the empirical Fourier coefficient

$$\widehat{p'}(S) = \frac{1}{2^n T} \sum_{s \in [T]} \prod_{i \in S} x_i^s.$$

Then, $\mathbb{E}[\widehat{p'}(S)] = \widehat{p}(S)$. Moreover, by a Hoeffding bound (Lemma A.2) and a union bound over the at most $n^k$ sets of size at most $k$, we have that with probability $\geq 1 - \delta$

$$|\widehat{p'}(S) - \widehat{p}(S)| \leq \frac{\varepsilon}{2 \cdot 2^n \sqrt{2^k}} \text{ for every } |S| \leq k. \quad (1)$$

For every $|S| \leq k$, we define

$$\widehat{p''}(S) = \begin{cases} 0 & \text{if } |\widehat{p'}(S)| \leq \varepsilon/(2 \cdot 2^n \cdot \sqrt{2^k}), \\ \widehat{p'}(S) & \text{otherwise.} \end{cases} \quad (2)$$

Now, from Equation (1) it follows that if $\widehat{p}(S) = 0$, then $\widehat{p''}(S) = 0$. In particular, suppose $K$ is the set of (at most $k$) variables that $p$ depends on. Then

$$\widehat{p''}(S) = 0, \text{ for every } S \not\subseteq K. \quad (3)$$

In addition, we have that for every $S$ with $|S| \leq k$

$$|\widehat{p''}(S) - \widehat{p}(S)| \leq \frac{\varepsilon}{2^n \sqrt{2^k}}. \quad (4)$$

We define $p''(x) = \sum_{|S| \leq k} \widehat{p''}(S) \prod_{i \in S} x_i$ and claim that is close to $p$. Indeed,

$$\begin{aligned}
\|p - p'\|_{L_2}^2 &= \sum_{S \subseteq K} |\widehat{p}(S) - \widehat{p''}(S)|^2 + \sum_{S \not\subseteq K} |\widehat{p''}(S)|^2 \\
&= \sum_{S \subseteq K} |\widehat{p}(S) - \widehat{p''}(S)|^2 \\
&\leq 2^k \frac{\varepsilon^2}{2^{2n} 2^k} = \frac{\varepsilon^2}{2^{2n}},
\end{aligned}$$

where in the first line we have used Parseval's identity; in the second line we have used Equation (3); and in the third Equation (4), and that there are $2^k$ subsets of a set with $k$ elements. Hence, $\|p - p'\|_{L_2} \leq \varepsilon/2^n$. Finally, as $\|\cdot\|_{\ell_1} \leq 2^n \|\cdot\|_{L_2}$, the result follows. $\square$

*Remark* 2.1. The algorithm described in the proof of Theorem 1.1 does not output a distribution, but only a function $p'\colon \{-1,1\}^n \to \mathbb{R}$ that is a $k$-junta. However, it is easy to round this $p'$ to a distribution in time $O(2^k)$ without harming the approximation, as we show in the appendix.

## 3. Learning and Testing Quantum Junta States

In this section, we prove Theorems 1.3 and 1.4. We begin by recalling the usual access model for quantum states (O'Donnell & Wright, 2016; Haah et al., 2017). We are given copies of $\rho^{\otimes m}$ for $m \in \mathbb{N}$ on which we can measure. We consider the trace distance

$$d_{\mathrm{tr}}(\rho, \rho') = \|\rho - \rho'\|_{\mathrm{tr}} \leq \mathrm{Tr}[|\rho - \rho'|].$$

Note that an $n$-qubit state $\rho$ is a $k$-junta state if and only if it can be written as

$$\rho = \sum_{\substack{P \in \{I,X,Y,Z\}^{\otimes n} \\ \mathrm{supp}(P) \subseteq K}} \widehat{\rho}(P) P,$$

for some $K \subseteq [n]$ of size $k$, where $\widehat{\rho}(P) = \mathrm{Tr}[\rho P]/2^n$ are the Pauli coefficients, $\mathrm{supp}(\otimes_{i \in [n]} P_i) = \{i \in [n]\colon P_i \neq I\}$. We emphasize that the quantum state is the generalization of classical probability distribution and that this generalization extends to the Pauli spectrum several notions related to the Fourier spectrum. Indeed, given a probability distribution $p\colon \{-1,1\}^n \to [0,1]$, it defines a $n$-qubit quantum state

$$\rho_p = \sum_{x \in \{-1,1\}^n} p(x) |x\rangle\langle x|,$$

that satisfies $\widehat{\rho_p}(P) = \widehat{p}(\mathrm{supp}(P))$ if $P \in \{I,Z\}^{\otimes n}$, and $\widehat{\rho_p}(P) = 0$ otherwise. Similarly, the biggest size of the

support of a $P$ such that $\widehat{\rho}(P) \neq 0$ generalizes the notion of degree. Furthermore, $p$ is a $k$-junta distribution if and only if $\rho_p$ is a $k$-junta state.

## 3.1. Learning junta states

As in the classical case, the non-zero Pauli coefficients of a $k$-junta state correspond to *low-degree* Pauli operators, those with small support, and they are at most $4^k$. Using this, we could learn $k$-junta states in a similar way that we used to learn $k$-junta distributions if we had a mechanism for learning the low-degree Pauli coefficients. Such a mechanism is the Classical Shadows algorithm by Huang, Kueng, and Preskill (Huang et al., 2020), which was later improved by Elben et al. in Sec.II.B. (Elben et al., 2022).

**Theorem 3.1.** *Let $\rho$ be a $n$-qubit state. Then, by performing Pauli measurements on $O(\frac{3^k \log((3n)^k/\delta)}{2^{2n}\varepsilon^2})$ single copies [4] of $\rho$, one can output estimates $\widehat{\rho}'(P)$ such that with success probability $\geq 1 - \delta$ satisfy*

$$|\widehat{\rho}(P) - \widehat{\rho}'(P)| \leq \varepsilon$$

*for every $P \in \{I, X, Y, Z\}^{\otimes n}$ with $|\mathrm{supp}(P)| \leq k$.*

In the appendix, we include a novel proof of Theorem 3.1 that uses a novel Pauli analytic approach inspired by the proof of the non-commutative Bohnenblust-Hille inequality by Volberg and Zhang (Volberg & Zhang, 2023). Notably, in contrast with the original proof, ours does not make use of the median of means estimator, just the empirical mean.

Our algorithm to learn $k$-junta states is robust, in the sense that it also applies in the case of the Pauli spectrum of the state is $(\varepsilon^2/2^{2n})$-concentrated on the Pauli coefficients corresponding to $k$-qubits, which is the case where it exists $K \subseteq [n]$ of size $k$ such that

$$\sum_{\mathrm{supp}(P) \not\subseteq K} |\widehat{\rho}(P)|^2 \leq \frac{\varepsilon^2}{2^{2n}}.$$

We will need this robustness for the application towards learning QAC$^0$ circuits. We also note that we could prove a robust version of Theorem 1.1, but we did not for simplicity.

**Theorem 3.2.** *Let $\rho$ be a $n$-qubit state whose Pauli spectrum is $(\varepsilon^2/2^{2n})$-concentrated on a set of $k$ qubits. Then, using $O(\frac{12^k \log((3n)^k/\delta)}{\varepsilon^2})$ copies of $\rho$ one can output $\rho'$ such that with success probability $\geq 1 - \delta$ satisfies*

$$\sum_{P \in \{I,X,Y,Z\}^{\otimes n}} |\widehat{\rho}'(P) - \widehat{\rho}(P)|^2 \leq \frac{\varepsilon^2}{2^{2n}}.$$

*In particular, $\|\rho' - \rho\|_{\mathrm{tr}} \leq \varepsilon$. Furthermore, the algorithm just does Pauli measurements on single copies of the state.*

---

[4]The unusual factor $2^{2n}$ appears because the Pauli coefficients are the expectations of the Pauli observables over $2^n$.

The proof of Theorem 3.2 follows the same strategy as the one for junta distributions, Theorem 1.1, so we defer it to the appendix.

The upper bound of Theorem 1.3 follows from Theorem 3.2, and the lower bound $\Omega(4^k/\varepsilon^2)$ follows from the fact that $k$-qubits states are $k$-juntas. To prove the lower bound $\Omega(\log(n)/\varepsilon^2)$, we find an ensemble of $n$ states that are 1-juntas and are difficult to distinguish (see Theorem C.1 for the details).

## 3.2. Testing quantum junta states

In this section, we prove Theorem 1.4. To do that, we introduce a proxy for the distance of $\rho$ to the space of $k$-junta states. The distance of $\rho$ to the space of $k$-junta state is

$$d(\rho, k\text{-junta}) := \inf_{K \subset [n], |K|=k, \sigma_K} \left\| \rho - \sigma_K \otimes \frac{I_{[n]-K}}{2^{n-k}} \right\|_{\mathrm{tr}},$$

where $\sigma_K$ is a state on the qubits indexed by $K$. The proxy is

$$\widetilde{d}(\rho, k\text{-junta}) := \inf_{K \subset [n], |K|=k} \left\| \rho - \rho_K \otimes \frac{I_{[n]-K}}{2^{n-k}} \right\|_{\mathrm{tr}}. \quad (5)$$

This proxy $\widetilde{d}$ is equivalent to $d$ up to constant factors, and it is easier to analyze.

**Proposition 3.3.** *Let $\rho$ be an $n$-qubit state and $k \in \mathbb{N}$. Then,*

$$d(\rho, k\text{-junta}) \leq \widetilde{d}(\rho, k\text{-junta}) \leq 2d(\rho, k\text{-junta}), \quad (6)$$

*where $d(\rho, k\text{-junta})$ is the minimum trace distance of $\rho$ to a $k$-junta state.*

*Proof.* The first inequality follows from the inclusion of the feasibility region of the infimum of $\widetilde{d}$ in the feasibility region of one of $d$. To prove the second inequality, consider a $k$-junta state $\sigma_K \otimes I_{[n]-K}/2^{n-k}$. By monotonicity of the trace norm we have that $\|\rho_K - \sigma_K\|_{\mathrm{tr}} \leq \|\sigma_K \otimes I_{[n]-K}/2^{n-k} - \rho\|_{\mathrm{tr}}$, so by triangle inequality it follows that

$$\|\rho_K \otimes I_{[n]-K} - \rho\|_{\mathrm{tr}} \leq 2\|\sigma_K \otimes I_{[n]-K}/2^{n-k} - \rho\|_{\mathrm{tr}}.$$

Now, the second inequality in the statement follows from taking infimums. $\square$

First, we will give the upper bound, in which we will use tomography (O'Donnell & Wright, 2016) and quantum state certification (Bădescu et al., 2019) as subroutines.

**Theorem 3.4** ((O'Donnell & Wright, 2016)). *Let $\rho$ be a $k$-qubit state. Then, $\Theta(4^k \log(1/\delta)/\varepsilon^2)$ copies of $\rho$ are necessary and sufficient to $\varepsilon$-learn $\rho$ in trace distance with success probability $\geq 1 - \delta$.*

**Theorem 3.5** ((Bǎdescu et al., 2019)). *Let $\rho$ be an unknown $n$-qubit state and $\rho'$ a known $n$-qubit state. Then, $\Theta(2^n \log(1/\delta)/\varepsilon^2)$ copies of $\rho$ are necessary and sufficient to test whether $\rho$ is $\varepsilon$-close or $2\varepsilon$-far in trace distance to $\rho'$ with success probability $\geq 1 - \delta$.*

We are ready to state our upper bound for quantum junta state testing.

**Theorem 3.6.** *Let $\rho$ be an $n$-qubit state. Then, $O(n^k 2^n \log(n^k/\delta)/\varepsilon^2)$ copies of $\rho$ are sufficient to test whether $\rho$ is $\varepsilon$-close or $7\varepsilon$-far in trace distance from being $k$-junta with success probability $\geq 1 - \delta$.*

*Proof.* First, we describe the algorithm. For every $K \subseteq [n]$ of size $k$, we perform the following procedure: $i$) we perform tomography on $K$ and obtain an estimate $\widetilde{\rho}_K$ such that $\|\widetilde{\rho}_K - \rho_K\|_{\mathrm{tr}} \leq \varepsilon$, $ii$) we test whether $\widetilde{\rho}_K \otimes I_{[n]-K}/2^{n-k}$ is $3\varepsilon$-close or $6\varepsilon$-far from $\rho$. If we find that any of the $\widetilde{\rho}_K \otimes I_{[n]-K}/2^{n-k}$ is close to $\rho$, we output that $\rho$ is close to a $k$-junta, and otherwise we output that it is far.

Second, we analyze the complexity. For every $K \subseteq [n]$ of size $k$, in order to succeed with probability $1 - \delta/n^k$ (so the total succeed probability is $\geq 1 - \delta$ by a union bound), the step $i$) uses $\Theta(4^k \log(n^k/\delta)/\varepsilon^2)$ copies of $\rho$, via Theorem 3.4, and the step $ii$) uses $O(2^n \log(n^k/\delta)/\varepsilon^2)$ copies, via Theorem 3.5. As the step $ii$) dominates (if $k < n/2$), and as there are at most $O(n^k)$ subsets of $[n]$ of size $k$, the total number of copies used are $O(n^k 2^n \log(n^k/\delta)/\varepsilon^2)$.

Finally, we analyze the correctness. If $\rho$ is $\varepsilon$-close to being a $k$-junta, then by Proposition 3.3 and triangle inequality, there exists $K$ such that

$$\left\|\widetilde{\rho}_K \otimes I_{[n]-k}/2^{n-k} - \rho\right\|_{\mathrm{tr}} \leq 3\varepsilon,$$

so the algorithm would output that $\rho$ is close to a $k$-junta, as desired. If $\rho$ is $7\varepsilon$-far from being a $k$-junta, then by Proposition 3.3 and triangle inequality follow that

$$\left\|\widetilde{\rho}_K \otimes I_{[n]-k}/2^{n-k} - \rho\right\|_{\mathrm{tr}} \geq 6\varepsilon,$$

for every $K \subseteq [n]$ of size $k$. Hence, the algorithm outputs that $\rho$ is far from junta, as desired. $\square$

We now focus on the lower bound for junta testing. To do that, we reduce the problem of testing whether an unknown state is the maximally mixed state to the problem of testing the maximally mixed state (O'Donnell & Wright, 2021).

**Theorem 3.7** ((O'Donnell & Wright, 2021)). *Let $\rho$ be an unknown $n$-qubit state. Then, $\Theta(2^n \log(1/\delta)/\varepsilon^2)$ copies of $\rho$ are necessary and sufficient to test whether $\rho$ is equal or $\varepsilon$-far in trace distance to the maximally mixed state with success probability $\geq 1 - \delta$.*

**Theorem 3.8.** *Let $\rho$ be an $n$-qubit state. Then, $\Omega(2^{n-k} \log(1/\delta)/\varepsilon^2)$ copies of $\rho$ are necessary to test whether $\rho$ is a $k$-junta or $\varepsilon$-far in trace distance from it with success probability $\geq 1 - \delta$.*

*Proof.* Assume that we had an algorithm $\mathcal{A}$ able to test whether $\rho$ is a $k$-junta state or $\varepsilon$-far from it. We further assume that $\rho$ equals $|0^k\rangle\langle 0^k| \otimes \rho'$ for a $(n-k)$-qubit state $\rho'$. We claim two things: $i$) if $\rho'$ is the maximally mixed state, then $\rho$ is a $k$-junta state; $ii$) if $\rho'$ is $2\varepsilon$-far from the maximally mixed state, then $\rho$ is $\varepsilon$-far from $k$-junta. Then, $\mathcal{A}$ would be able to test whether the $(n-k)$-qubit state $\rho'$ is maximally mixed or $2\varepsilon$-far from it, so by Theorem 3.7 follows that $\mathcal{A}$ uses $\Omega(2^{n-k} \log(1/\delta)/\varepsilon^2)$ copies of $\rho$.

We now prove $i$) and $ii$). $i$) follows from the definition. To prove $ii$), let $K \subset [n]$ of size $k$. If $K = [k]$, then by monotonicity of the trace norm

$$\left\|\rho_K \otimes \frac{I_{[n]-K}}{2^{n-k}} - \rho\right\|_{\mathrm{tr}} \geq \left\|\frac{I_{[n]-K}}{2^{n-k}} - \rho'\right\|_{\mathrm{tr}} \geq 2\varepsilon. \quad (7)$$

If $K \neq [k]$, then there is $i \in [k]$ such that $i \notin K$, so again by monotonocity of the trace norm

$$\left\|\rho_K \otimes \frac{I_{[n]-K}}{2^{n-k}} - \rho\right\|_{\mathrm{tr}} \geq \left\|\frac{I_{\{i\}}}{2} - |0\rangle\langle 0|\right\|_{\mathrm{tr}} = 1. \quad (8)$$

Finally, $ii$) follows from Equations (7) and (8) and Proposition 3.3. $\square$

*Proof of Theorem 1.4.* The proof follows from Theorems 3.6 and 3.8. $\square$

# 4. QAC0 Circuits

## 4.1. Very brief introduction to QAC0 circuits

A QAC0 is a circuit composed of single-qubit gates and Toffoli gates, which are the unitaries defined via

$$|x_1, \ldots, x_l, b\rangle \mapsto |x_1, \ldots, x_l, b \cdot \mathsf{AND}(x_1, \ldots, x_l)\rangle,$$

where here $x_1, \ldots, x_l, b \in \{-1, 1\}$ and $\mathsf{AND}(x_1, \ldots, x_l) = -1$ if and only if $x_1 = \cdots = x_l = -1$. Given a $(n + a + 1)$-qubit QAC0 circuit, one should think of the first $n$ qubits as input qubits, of the next $a$ qubits as auxiliary qubits, and of the last qubit as an output qubit. Also, the last $(a + 1)$ qubits are initialized in a fixed state $\sigma$. Hence, a QAC0 circuit defines an $n$-to-1 qubit channel via

$$\Phi_\sigma(\rho) = \mathrm{Tr}_{[n+a]}\left[U(\rho \otimes \sigma)U^\dagger\right],$$

where $U$ is the unitary implemented by the circuit and $\mathrm{Tr}_{[n+a]}$ is the trace with respect to the input and auxiliary

qubits. The Choi state of a $\mathsf{QAC}^0$ circuit is the Choi state of its correspondent channel, namely the $(n+1)$-qubit state

$$\rho_{\Phi_\sigma} = \Phi_\sigma \otimes I_n(|\mathrm{EPR}_n\rangle\langle\mathrm{EPR}_n|),$$

where $|\mathrm{EPR}_n\rangle$ is the tensor product of $n$ EPR states.

The original motivation when Moore introduced of $\mathsf{QAC}^0$ circuits was to use them to approximate Boolean functions $f\colon \{-1,1\}^n \to \{-1,1\}$ (Moore, 1999), namely to approximate $n$-to-1 qubit channels like

$$\Phi_f(\rho) = \sum_{x\in\{-1.1\}^n} \langle x|\rho|x\rangle |f(x)\rangle\langle f(x)|.$$

It is easy to check that the Choi state of these channels is given by

$$\rho_f = \frac{1}{2^n}\left(I^{\otimes(n+1)} + \sum_{S\subseteq[n]} \widehat{f}(S)Z_S \otimes Z\right), \quad (9)$$

where $Z_S = \otimes_{i\in[n]}Z^{\delta_{i\in S}}$. Hence, for $f,g\colon \{-1,1\}^n \to \{-1,1\}$, we have that

$$2^n|\rho_f - \rho_g|_F^2 = 2^{2n} \sum_{P\in\{I,X,Y,Z\}^{\otimes(n+1)}} |\widehat{\rho}_f(P) - \widehat{\rho}_g(P)|^2$$

$$= \sum_{S\subseteq[n]} \left|\widehat{f}(S) - \widehat{g}(S)\right|^2 = \Pr[f(x)\neq g(x)], \quad (10)$$

where in the first equality we have used Parseval's identity, in the second Equation (9) and the last equality is elementary. From Equation (9) follows that learning the Choi state of a $\mathsf{QAC}^0$ circuit in the $2^n$-Frobenius norm is a pretty natural problem.

### 4.2. $\mathsf{QAC}^0$ circuits are close to juntas

Nadimpalli et al. showed that, for fixed depth and size, the Pauli spectrum of the Choi state of a $\mathsf{QAC}^0$ circuit is concentrated on low-degree coefficients by Theorem 18 in (Nadimpalli et al., 2024). However, we notice that something stronger holds, namely that these states are close to being juntas.

**Theorem 4.1.** *Let $\rho$ be the Choi state of $(n+a+1)$ $\mathsf{QAC}^0$ circuit of depth $d$ and size $s$ and let $\varepsilon > 0$. Then, there exists a set $K\subseteq[n+1]$ with $|K|\leq \left(\log\left(2^a s^2/\varepsilon\right)\right)^d$ such that*

$$\sum_{\mathrm{supp}(P)\not\subseteq K} |\widehat{\rho}(P)|^2 \leq \frac{\varepsilon}{2^{2n}}.$$

To prove Theorem 4.1, we have to borrow a few lemmas from (Nadimpalli et al., 2024) and apply them in a careful way. In that work, results are stated for the Choi representation of a channel, and in our work, we use the Choi state of

the channel. Both are easily related, as the Choi state is obtained by dividing the Choi representation by the dimension of the space.

Let $U$ be the unitary implemented by a $(n+a+1)$ $\mathsf{QAC}^0$ circuit. Then, it defines a $(n+a+1)$-to-1 qubit channel via

$$\Phi(\cdot) = \mathrm{Tr}_{[n+a]}\left[U\cdot U^\dagger\right].$$

The first lemma we need states that the Choi state of this $(n+a+1)$-to-1 qubit channel does not change much if one removes from the circuit a few Toffoli gates acting on many qubits as Lemma 23 in (Nadimpalli et al., 2024).

**Lemma 4.2** ((Nadimpalli et al., 2024)). *Let $\Phi$ be the $(n+a+1)$-to-1 channel defined by an $(n+a+1)$-qubit $\mathsf{QAC}^0$ circuit. Let $\Phi'$ be the $(n+a+1)$-to-1 channel obtained by removing from the circuit $m$ Toffoli gates acting on at least $l$ qubits each. Then, the Choi states satisfy*

$$\sum_P |\widehat{\rho}_\Phi(P) - \widehat{\rho}_{\Phi'}(P)|^2 = O\left(\frac{m^2}{2^l 2^{2(n+a+2)}}\right).$$

Recall that $(n+a+1)$-qubit $\mathsf{QAC}^0$ circuit also defines an $n$-to-1 qubit channel when the auxiliary register is initialized on a fixed $(a+1)$-qubit state $\sigma$, namely $\Phi_\sigma(\rho\otimes\sigma) = \Phi(\rho\otimes\sigma)$. The second lemma we need relates to the Pauli spectrum of the Choi states of $\Phi_\sigma$ and $\Phi$ as Proposition 28 in (Nadimpalli et al., 2024).

**Lemma 4.3** ((Nadimpalli et al., 2024)). *Let $\Phi$ and $\Phi_\sigma$ be the channels as above determined by a $\mathsf{QAC}$ circuit. Then, their Choi states satisfy*

$$\widehat{\rho}_{\Phi_\sigma}(P) = 2^{a+1} \sum_{Q\in\{I,X,Y,Z\}^{\otimes n}} \widehat{\rho}_\Phi(P\otimes Q)\,\mathrm{Tr}\left[Q\sigma^T\right].$$

*Proof of Theorem 4.1.* Let $\Phi$ be the $(n+a+1)$-to-1 channel determined by $(n+a+1)$-qubit $\mathsf{QAC}^0$ circuit of depth $d$ a size $s$. Let $l\in\mathbb{N}$ be fixed later. Let $\Phi'$ be the $(n+a+1)$-to-1 channel obtained by removing from the circuit the Toffoli gates that act on more than $l$ qubits. As there are at most $s$ of them, by Lemma 4.2 we have that the Choi states satisfy

$$\sum_P |\widehat{\rho}_\Phi(P) - \widehat{\rho}_{\Phi'}(P)|^2 = O\left(\frac{s^2}{2^l 2^{2(n+a+1)}}\right). \quad (11)$$

Now, by a light-cone argument, as the depth of the circuit without the *long* Toffoli gates is at most $d$, then at the end of the circuit the output qubit only depends on at most $l^d$ other qubits. This implies that the $\rho_{\Phi'}$ is a $k$-junta state for $k = l^d + 1$. By Equation (11), if $K\subseteq[n+a+2]$ is the set of $k$ qubits on which $\rho_{\Phi'}$ depends on, then

$$\sum_{P\notin\{I,X,Y,Z\}^K} |\widehat{\rho}_\Phi(P)|^2 = O\left(\frac{s^2}{2^l 2^{2(n+a+2)}}\right). \quad (12)$$

Now, if $K' \subseteq [n+1]$ is the subset of non-auxiliary qubits of $K$, i.e., $K' = K \cap [n+1]$, then

$$\sum_{\text{supp}(P) \subseteq [K']} |\widehat{\rho}_{\Phi'}(P)|^2$$

$$= 2^{2(a+1)} \sum_{\text{supp}(P) \subset [K']} \left| \sum_Q \widehat{\rho}_\Phi(P \otimes Q) \text{Tr}[Q\sigma^T] \right|^2$$

$$\leq 2^{2(a+1)} \sum_P \left( \sum_Q |\widehat{\rho}_\Phi(P \otimes Q)|^2 \right) \cdot \left( \sum_Q |\text{Tr}[Q\sigma^T]|^2 \right)$$

$$= 2^{3(a+1)} \|\sigma^T\|_F^2 \sum_{\text{supp}(P) \subset [K']} \sum_Q |\widehat{\rho}_\Phi(P \otimes Q)|^2$$

$$\leq 2^{3(a+1)} \sum_{\text{supp}(P) \not\subseteq [K]} |\widehat{\rho}_\Phi(P)|^2$$

$$= 2^{3(a+1)} O\left( \frac{s^2}{2^l 2^{2(n+a+2)}} \right) = O\left( \frac{s^2 2^{a+1}}{2^l 2^{2(n+1)}} \right),$$

where the range of the sums over $Q$ is $\{I, X, Y, Z\}^{\otimes(a+1)}$; the first line is true by Lemma 4.3; in the second we apply Cauchy-Schwarz inequality; in the third we use Parseval identity with $\sigma^T$; in the fourth line we use that if $\text{supp}(P) \not\subseteq K'$, then $\text{supp}(P \otimes Q) \not\subseteq K$; and in the fifth line we use Equation (12). Now, the result follows by taking $l = \log(s^2 2^{a+1}/\varepsilon^2)$. $\qquad\square$

## 4.3. Learning QAC0 circuits

In this section, we prove Theorem 1.5, which we restate for the reader's convenience.

**Theorem 1.5.** *Let $\rho$ be the Choi state of a $n$-qubit QAC0 circuit with size $s$, depth $d$, and $a$ auxiliary qubits. Then using $2^{O((\log(s^2 2^a/\varepsilon))^d)} \log(n/\delta)$ copies of $\rho$, one can output a $\rho'$ such that with probability at least $1 - \delta$ satisfies*

$$2^n \|\rho - \rho'\|_F^2 \leq \varepsilon.$$

*Furthermore, the algorithm just does Pauli measurements on single copies of the state.*

*Proof.* Theorem 1.5 quickly follows from Theorems 3.2 and 4.1 and using that for two $n$-qubit states $\rho$ and $\rho'$ we have that by Parseval's identity

$$2^{2n} \sum_{P \in \{I,X,Y,Z\}^{\otimes n}} |\widehat{\rho}(P) - \widehat{\rho'}(P)|^2 = 2^n \|\rho - \rho'\|_F^2.$$

$\qquad\square$

## 4.4. New lower bounds on the computing power of QAC0 circuits

Finally, we show how to use Theorem 4.1 to improve on the lower bounds on the computing power of QAC0 circuits. To

improve on the lower bounds that one would obtain with Theorem 18 from (Nadimpalli et al., 2024), one should look for functions of low degree that are far from being juntas. With that purpose, we consider the address function, which is known to be the Boolean function of degree $D+1$ that depends on more variables (Nisan & Szegedy, 1994). To define it, let add : $\{-1,1\}^D \to [2^D]$ be a bijection. The $D$-address function $f: \{-1,1\}^D \times \{-1,1\}^{2^D} \to \{-1,1\}$ is defined by

$$f(x,y) = \sum_{a \in \{-1,1\}^D} \left( \frac{x_1 a_1 + 1}{2} \right) \cdots \left( \frac{x_k a_k + 1}{2} \right) y_{\text{add}(a)}$$

for every $x \in \{-1,1\}^D$ and $y \in \{-1,1\}^{2^D}$. Note that $f$ has degree $D+1$, but depends on $2^D + D$ variables. Moreover, we can show that $f$ is far from every Boolean function that depends on fewer than $2^D$ variables.

*Fact* 4.4. Let $f$ be the $D$-address function. Let $k \in [2^D]$. Then, the degree of $f$ is $D+1$ and $f$ is $((2^D-k)/2^{D+1})$-far from being a $k$-junta.

*Proof.* Let $g: \{-1,1\}^{D+2^D} \to \{-1,1\}$ be a $k$-junta. The distance $d(f,g)$ between $g$ and $f$ is

$$\Pr_{x,y}[g(x,y) \neq f(x,y)] = 1 - \Pr_{x,y}[g(x,y) = y_{\text{add}(x)}],$$

where in the last equality we have used that $(\frac{x_1 a_1+1}{2}) \cdots (\frac{x_k a_k+1}{2}) = \delta_{a,x}$. Now,

$$\Pr_{x,y}[g(x,y) = y_{\text{add}(x)}] = \frac{1}{2^D} \sum_x \Pr_y[g(x,y) = y_{\text{add}(x)}]$$

$$\leq \frac{1}{2^D} \left( k + \frac{1}{2}(2^D - k) \right),$$

where in the inequality we have used that $g$ depends on at most $k$ variables of $y_1, \ldots, y_{2^D}$, and that if $g$ does not depend on $y$, then $\Pr_y[g(x,y) = y_{\text{add}(x)}] = 1/2$. Putting everything together follows that $d(f,g) \geq ((2^D - k)/2^{D+1})$. $\qquad\square$

**Corollary 4.5.** *In order to compute the $D$-address function with a depth $d$, size $s$ QAC0 circuit with $a$-auxiliary qubits up to error 1/4, the parameters need to satisfy*

$$s^2 2^a = \Omega\left( 2^{(2^D)^{1/d}} \right).$$

*Proof.* By Fact 4.4 it follows that the $D$-address function is $1/8$-far from every $((3/4)2^D)$-junta. On the other hand, by Theorem 4.1 it follows that the Choi-state of the QAC0 that does not satisfy $\log(s^2 2^a)^d = \Omega(2^D)$ circuit is $1/8$-close to a $((3/4)2^D)$-junta. Putting both things together, the claimed result follows. $\qquad\square$

*Remark* 4.6. If one used Theorem 18 in (Nadimpalli et al., 2024) instead of Theorem 4.1 in the proof of Corollary 4.5, one would obtain a weaker lower bound $s^2 2^a = \Omega(2^{D/d})$.

## Acknowledgments

We thank Jop Briët, Alexandros Eskenazis, Yuval Filmus, Jonas Helsen, Dale Jacobs, and Francisca Vasconcelos for useful conversations. J.B. was supported by the Quantum Advantage Pathfinder project of UKRI Engineering and Physical Sciences Research Council under grant No. EP/X026167/1. Partial work was done when J.B. was in the Centre for Quantum Technologies, National University of Singapore, supported under grant No. A-0009870-00-00. F.E.G. thanks the Hausdorff Research Institute of Mathematics of Bonn, which hosted F.E.G. during the Dual Trimester Program: "Boolean Analysis in Computer Science", where part of this paper was written. F.E.G was supported by the European Union's Horizon 2020 research and innovation programme under the Marie Skłodowska-Curie grant agreement No. 945045, and by the NWO Gravitation project NETWORKS under grant No. 024.002.003. We thank the anonymous referees for their helpful feedback.

## Impact Statement

This paper presents work whose goal is to advance the field of machine learning. There are many potential societal consequences of our work, none of which we feel must be specifically highlighted here. Furthermore, the work is about the most theoretical aspects of machine learning, namely computational learning theory, so no undesirable societal consequences are to be expected in the short term.

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

# A. Preliminaries

We use $\ell_q$ to denote the $q$-norm with the counting measure, and $L_q$ to denote the $q$-norm with the uniform probability measure. All expectations are taken with respect to the uniform probability measure unless otherwise stated. All logarithms are in base 2. We denote the Frobenius norm of a matrix $M$ by $\|M\|_{\mathrm{F}} = \sqrt{\mathrm{Tr}[M^\dagger M]}$, and the trace norm by $\|M\|_{\mathrm{Tr}} = \mathrm{Tr}[\sqrt{M^\dagger M}] = \|M\|_{S_1}$. For $n \geq 1$, we write $[n] = \{1, \dots, n\}$. We write $I_{[n]}$ to denote the $2^n \times 2^n$ identity matrix. Given $S \subseteq [n]$, we write $I_S$ to denote the identity $2^{|S|} \times 2^{|S|}$ identity matrix acting on the qubits indexed by $S$.

## A.1. Brief introduction to quantum information

For an extensive introduction to quantum information, we refer the reader to (Nielsen & Chuang, 2010; Wilde, 2013; Watrous, 2018). A quantum state on $n$ qubits is a positive semidefinite $2^n \times 2^n$ complex matrix with trace 1. In particular, a probability distribution on $n$ bits defines an $n$-qubit quantum state by embedding the values of the probability in the diagonal of a matrix. Quantum systems are described by states, and the way to extract information from them is via the outcome of measurements. Formally, a measurement on $n$-qubits is a family of positive semidefinite $2^n \times 2^n$ complex matrices that sum to the identity. If $\mathcal{X}$ is an alphabet, and $\{M_x\}_{x \in \mathcal{X}}$ is a measurement, the probability of outputting the outcome $x$ when measuring a state $\rho$ is given by $\mathrm{Tr}[\rho M_x]$.

The standard way of showing lower bounds for quantum state learning is via an argument of Holevo. To state it, we first introduce the Holevo information of a set of states $\{\rho_i\}$, which is given by

$$\chi(\{\rho_i\}) = S\left(\frac{1}{n} \sum_{i \in [n]} \rho_i\right) - \frac{1}{n} \sum_{i \in [n]} S(\rho_i),$$

where $S(\rho) = -\mathrm{Tr}[\rho \log(\rho)]$ is the von Neumann entropy. Now we are ready to write the precise statement we will use to show a lower bound for learning $k$-junta states. For a proof, see Lemma S14 in (Mele et al., 2025).

**Lemma A.1.** *Let $\{\rho_i\}_{i \in [M]}$ be a family of $M$ states that satisfy $\|\rho_i - \rho_j\|_{\mathrm{Tr}} \geq \varepsilon$ for every $i \neq j$. Assume that $T$ copies are sufficient to learn this family of states with probability $\geq 2/3$. Then,*

$$\chi(\{\rho_i^{\otimes T}\}) = \Omega(\log(M)).$$

Additionally, we will need two facts about von Neumann entropy. The first is its additivity under the tensor product,

$$S(\rho_A \otimes \rho_B) = S(\rho_A) + S(\rho_B), \tag{13}$$

and the second is subadditivity

$$S(\rho_{AB}) \leq S(\rho_A) + S(\rho_B). \tag{14}$$

## A.2. Fourier and Pauli analysis

**Fourier analysis.** In this section, we will talk about the space of functions defined on the Boolean hypercube $f \colon \{-1, 1\}^n \to \mathbb{R}$ endowed with the inner product $\langle f, g \rangle = \mathbb{E}_x[f(x)g(x)]$, where the expectation is taken with respect to the uniform measure of probability. For $S \subseteq [n]$, the Fourier characters, defined by $\prod_{i \in S} x_i$, constitute an orthonormal basis of this space. Hence, every $f$ can be identified with a multilinear polynomial via the Fourier expansion

$$f = \sum_{S \subseteq [n]} \widehat{f}(s) \prod_{i \in S} x_i,$$

where $\widehat{f}(S)$ are the Fourier coefficients given by $\widehat{f}(S) = \mathbb{E}_x[f(x) \prod_{i \in S} x_i]$. The degree of $f$ is the minimum $d$ such that $\widehat{f}(S) = 0$ if $|S| > d$. We will often use Parseval's identity:

$$\|f\|_{L_2}^2 = \langle f, f \rangle = \sum_{S \subseteq [n]} \widehat{f}(S)^2.$$

For an extensive introduction to Fourier analysis, see (O'Donnell, 2014).

**Pauli analysis.** In this section we consider the space of $2^n \times 2^n$ complex matrices endowed with the usual inner product $\langle A, B \rangle = \frac{1}{2^n} \operatorname{Tr}[A^\dagger B]$. The Pauli operators $\{I, X, Y, Z\}^{\otimes n}$, where

$$I = \begin{pmatrix} 1 & 0 \\ 0 & 1 \end{pmatrix}, \quad X = \begin{pmatrix} 0 & 1 \\ 1 & 0 \end{pmatrix}, \quad Y = \begin{pmatrix} 0 & -i \\ i & 0 \end{pmatrix}, \quad \text{and} \quad Z = \begin{pmatrix} 1 & 0 \\ 0 & -1 \end{pmatrix},$$

form an orthonormal basis for this space. The Pauli expansion of a matrix $M$ is given by

$$M = \sum_{P \in \{I, X, Y, Z\}^{\otimes n}} \widehat{M}(P) P, \tag{15}$$

where $\widehat{M}(P) = \langle P, M \rangle$ are Pauli coefficients of $M$. We will refer to the collection of non-zero Pauli coefficients as the Pauli spectrum of $M$. As $\{P\}_{P \in \{I, X, Y, Z\}^{\otimes n}}$ is an orthonormal basis, we have Parseval's identity

$$\langle M, M \rangle = \sum_{P \in \{I, X, Y, Z\}^{\otimes n}} \left| \widehat{M}(P) \right|^2.$$

Given a matrix $M$, its degree is the minimum $d$ such that $\widehat{M}(P) = 0$ for any $P$ that takes more than $d$ times a non-identity value. This notion of degree for matrices generalizes the classical notion of Fourier degree. For an extensive introduction to Pauli analysis, see (Montanaro & Osborne, 2010).

### A.3. Concentration inequalities

We state a few concentration inequalities that we often use.

**Lemma A.2** (Hoeffding bound). *Let $X_1, \ldots, X_m$ be independent-random variables that satisfy $-a_i \leq |X_i| \leq a_i$ for some $a_i > 0$. Then, for any $\tau > 0$, we have*

$$\Pr \left[ \left| \sum_{i \in [m]} X_i - \sum_{i \in [m]} \mathbb{E}[X_i] \right| > \tau \right] \leq 2 e^{-\frac{\tau^2}{2(a_1^2 + \cdots + a_m^2)}}.$$

**Lemma A.3** (Bernstein inequality). *Let $X_1, \ldots, X_m$ be independent-random variables with $|X_i| \leq M$ for some $M > 0$. Then,*

$$\Pr \left[ \left| \sum_{i \in [m]} X_i - \sum_{i \in [m]} \mathbb{E}[X_i] \right| > \tau \right] \leq 2 e^{-\frac{\tau^2/2}{\sum_{i \in [m]} \operatorname{Var}[X_i] + \tau M/3}}.$$

## B. Our learning algorithms in a nutshell

All of our algorithms are refinements of the *low-degree algorithm* of Linial, Mansour, and Nisan (Linial et al., 1993). To sketch them, for simplicity, we will consider functions $f \colon \{-1, 1\}^n \to [-1, 1]$. Assume that we are promised that the Fourier spectrum of $f$ is supported on $L$ monomials of degree at most $d$, i.e.

$$f(x) = \sum_{s \in [L]} \widehat{f}(S_s) \prod_{i \in S_s} x_i$$

for some $S_s \subseteq [n]$ with $|S_s| \leq d$. First, we will see how the low-degree algorithm would perform to learn $f$ from samples $(x, f(x))$ where $x$ is uniformly picked from $\{-1, 1\}^n$.

---

**Low-degree algorithm**

**Step 1.** For every $|S| \leq d$, obtain $\widehat{f'}(S)$ that approximates $\widehat{f}(S)$ up to error $\sqrt{\varepsilon/n^d}$.

**Output.** We output $f'(x) = \sum_{|S| \leq d} \widehat{f'}(S) \prod_{i \in S} x_i$.

---

It is well-known that with $(1/\alpha^2) \cdot \log(M)$ samples one can estimate $M$ Fourier coefficients of $f$ up to error $\alpha$, so the low-degree algorithm requires $(n^d/\varepsilon^2) \cdot \log(n^d)$ samples. Now, $f'$ is close to $f$, because

$$\sum_{|S| \leq d} \left| \widehat{f}(S) - \widehat{f'}(S) \right|^2 \leq \sum_{|S| \leq d} \frac{\varepsilon}{n^d} \leq \varepsilon,$$

where in the first inequality we have used the guarantees of Step 1, and in the second that $|\{|S| \leq d\}| \leq n^d$. In particular, this implies that $\Pr[f(x) \neq \mathrm{sign}(f'(x))] \leq \varepsilon$.

However, note that the low-degree algorithm does not use the fact that $f$ is supported on $L$ monomials out of the $\sim n^d$ low-degree monomials. Using that, one can improve on the low-degree algorithm.

---

**Low-degree and sparse algorithm**

**Step 1.** For every $|S| \leq d$, obtain $\widehat{f'}(S)$ that approximates $\widehat{f}(S)$ up to error $\sqrt{\varepsilon/4L}$.

**Step 2.** For every $|S| \leq d$, if $|\widehat{f'}(S)| \leq \sqrt{\varepsilon/4L}$, set $\widehat{f'}(S) = 0$, otherwise set $\widehat{f''}(S) = \widehat{f'}(S)$.

**Output.** We output $f''(x) = \sum_{|S| \leq d} \widehat{f''}(S) \prod_{i \in S} x_i$.

---

Note that Step 1 now just requires $(L/\varepsilon^2) \cdot \log(n^d)$ samples, considerably less than the $(n^d/\varepsilon^2) \cdot \log(n^d)$ samples of the low-degree algorithm. Also, notice that by adding the rounding of Step 2 we make sure that $\widehat{f''}(S) = 0$ for $S \notin \{S_1, \dots, S_L\}$ and every $S \in \{S_1, \dots, S_L\}$ satisfies that $|\widehat{f}(S) - \widehat{f''}(S)| \leq \sqrt{\varepsilon/L}$. Hence, we still have that

$$\sum_{|S| \leq d} \left| \widehat{f}(S) - \widehat{f'}(S) \right|^2 = \sum_{S \in \{S_1, \dots, S_L\}} \left| \widehat{f}(S) - \widehat{f'}(S) \right|^2 \leq \sum_{S \in \{S_1, \dots, S_L\}} \frac{\varepsilon}{L} = \varepsilon,$$

where in the first equality we have used that $\widehat{f''}(S) = 0$ for every $S \notin \{S_1, \dots, S_L\}$.

In conclusion, if we are promised that the Fourier or Pauli spectrum of our object is supported on $L \ll n^d$ coefficients of degree at most $d$, one should add a rounding step in the low-degree algorithm. This remark is used by Eskenazis, Ivanisvili and Streck to learn Boolean functions (Eskenazis et al., 2023). In this work, we generalize it into broader cases, especially for learning quantum objects.

## C. Deferred proofs

Here, we will include the proofs of some of our theorems, which we will restate for convenience.

### C.1. Learning junta distributions

In this section, we show that, as noted in Remark 2.1, the algorithm described in the proof of Theorem 1.1 can be modified to output a junta distribution, without harming the approximation and with post-processing time $O(2^k)$. Indeed, that algorithm outputs a function $p' \colon \{-1, 1\}^n \to \mathbb{R}$ that is a $k$-junta. Let $K \subseteq [n]$ be the set of $k$ variables that $p'$ depends on. We consider $p'|_K$ as the restriction of $p'$ to these variables and round all its values to 0 if they are negative. This step only takes time $O(2^k)$, and it does not harm the approximation because the range of $p$ is contained in $[0, \infty)$. Let $p''|_K \colon \{-1, 1\}^k \to [0, \infty)$ the resulting function and $p'' \colon \{-1, 1\}^n \to [0, \infty) \colon x \to p''|_K(x_K)$ its extension to $n$ variables. We define $C = 2^{n-k} \sum_{x \in \{-1,1\}^k} p'|_K(x)$, which we can compute in time $O(2^k)$. As $\|p - p''\|_{\ell_1} \leq \varepsilon$, by triangle inequality we have that

$$|1 - C| = \left| \sum_{x \in \{-1,1\}^n} p(x) - \sum_{x \in \{-1,1\}^n} p''(x) \right| \leq \|p - p''\|_{\ell_1} \leq \varepsilon.$$

We now define $p''' := p''(x)/C$. By construction, $p'''$ is a probability distribution. In addition, $p'''$ is $O(\varepsilon)$-close to $p$, because

$$\|p''' - p\|_{\ell_1} \leq \|p''' - p''\|_{\ell_1} + \|p'' - p\|_{\ell_1} \leq \left(1 - \frac{1}{C}\right) + \varepsilon \leq O(\varepsilon),$$

where in the second step we have used $C \approx 1 \pm \varepsilon$ and that for $\varepsilon = O(1)$, $|1/(1 \pm \varepsilon) - 1| = O(\varepsilon)$ by Taylor's theorem. Hence, the total time complexity of the algorithm is $O(n^k \cdot 2^k k \log(n)/\varepsilon^2)$, coming from computing the $O(n^k)$ empirical low-degree Fourier coefficients.

## C.2. Learning and testing junta states

**Theorem 3.1.** *Let $\rho$ be a $n$-qubit state. Then, by performing Pauli measurements on $O(\frac{3^k \log((3n)^k/\delta)}{2^{2n}\varepsilon^2})$ single copies [5] of $\rho$, one can output estimates $\widehat{\rho}'(P)$ such that with success probability $\geq 1 - \delta$ satisfy*

$$|\widehat{\rho}(P) - \widehat{\rho}'(P)| \leq \varepsilon$$

*for every $P \in \{I, X, Y, Z\}^{\otimes n}$ with $|\mathrm{supp}(P)| \leq k$.*

*Proof.* We will make use of $T = O(3^k \log((3n)^k/\delta)/(2^{2n}\varepsilon^2))$ copies of $\rho$. Let $B_Q$ be a basis that diagonalizes $Q \in \{X, Y, Z\}^{\otimes n}$. For every $s \in [T]$, we will pick $Q^s \in \{X, Y, Z\}^{\otimes n}$ independently uniformly at random and measure $\rho$ in the basis $B_{Q^s}$. For every $i \in [n]$, let $x_i^s = \pm 1$ if the outcome of the $s$-th measurement on the $i$-th qubit is the $\pm 1$ eigen-space of $Q_i^s$. Then, for every $P \in \{I, X, Y, Z\}^{\otimes n}$ we define a empirical estimator of $\widehat{\rho}(P)$ via

$$\widehat{\rho}'(P) = \frac{3^{|\mathrm{supp}(P)|}}{2^n T} \sum_{s \in [T]} \prod_{i \in \mathrm{supp}(P)} x_i^s \delta_{P_i = Q_i^s}.$$

We claim that $\widehat{\rho}'(P)$ equals $\widehat{\rho}(P)$ on expectation. Indeed,

$$\mathbb{E}[\widehat{\rho}'(P)] = \frac{3^{|\mathrm{supp}(P)|}}{2^n} \mathbb{E}_{Q \in \{X,Y,Z\}^{\otimes n}} \prod_{i \in \mathrm{supp}(P)} \sum_{x_i \in \{-1,1\}} \Pr_{\rho, B_{Q_i}}[x_i] x_i \delta_{P_i = Q_i}$$

$$= \frac{3^{|\mathrm{supp}(P)|}}{2^n} \mathbb{E}_{Q \in \{X,Y,Z\}^{\otimes \mathrm{supp}(P)}} \prod_{i \in \mathrm{supp}(P)} \sum_{x_i \in \{-1,1\}} x_i \Pr_{\rho, B_{Q_i}}[x_i] \delta_{P_i = Q_i}$$

$$= \frac{1}{2^n} \prod_{i \in \mathrm{supp}(P)} \sum_{x_i \in \{-1,1\}} x_i \Pr_{\rho, P_i}[x_i]$$

$$= \frac{1}{2^n} \prod_{i \in \mathrm{supp}(P)} \mathrm{Tr}[\rho P_i]$$

$$= \frac{1}{2^n} \mathrm{Tr}[\rho P]$$

$$= \widehat{\rho}(P),$$

the first line is true because the expectation of $\widehat{\rho}'(P)$ does not change if $T$ changes; the second line follows from the fact that inside $\mathbb{E}_Q$ there is no dependence on the variables outside $\mathrm{supp}(P)$; the third line is true because the term inside $\mathbb{E}_Q$ is 0 unless $Q_i = P_i$ for every $i \in \mathrm{supp}(P)$; and fourth line is true because $\Pr_{\rho, B_{P_i}}[x_i] = \mathrm{Tr}[\rho|P_i(x_i)\rangle\langle P_i(x_i)|]$ where $|P_i(x_i)\rangle$ is a unit eigenvector of $P_i$ with eigenvalue $x_i$.

In addition, the second moment (and thus the variance) of $\widehat{\rho}'(P)$ for $T = 1$ is considerably smaller than the trivial upper bound $\mathbb{E}[|\widehat{\rho}'(P)|^2] \leq \|\widehat{\rho}'(P)\|_\infty^2 = 9^{|\mathrm{supp}(P)|}/4^n$. Indeed, for $T = 1$ we have

$$|\mathbb{E}[\widehat{\rho}'(P)]|^2 = \frac{9^{|\mathrm{supp}(P)|}}{4^n} \mathbb{E}_{Q \in \{X,Y,Z\}^{\otimes n}} \prod_{i \in \mathrm{supp}(P)} \sum_{x_i \in \{-1,1\}} \Pr_{\rho, B_{Q_i}}[x_i](x_i \delta_{P_i = Q_i})^2$$

$$= \frac{9^{|\mathrm{supp}(P)|}}{4^n} \mathbb{E}_{Q \in \{X,Y,Z\}^{\otimes \mathrm{supp}(P)}} \prod_{i \in \mathrm{supp}(P)} \sum_{x_i \in \{-1,1\}} \Pr_{\rho, B_{Q_i}}[x_i] \delta_{P_i = Q_i}$$

$$= \frac{9^{|\mathrm{supp}(P)|}}{4^n} \mathbb{E}_{Q \in \{X,Y,Z\}^{\otimes \mathrm{supp}(P)}} \prod_{i \in \mathrm{supp}(P)} \delta_{P_i = Q_i}$$

$$= \frac{3^{|\mathrm{supp}(P)|}}{4^n},$$

---

[5]The unusual factor $2^{2n}$ appears because the Pauli coefficients are the expectations of the Pauli observables over $2^n$.

where the second line follows from the fact that the quantity inside $\mathbb{E}_{Q \in \{X,Y,Z\}^{\otimes n}}$ does not depend on the variables outside of $\mathrm{supp}(P)$ and the fact that $(x_i \delta_{P_i = Q_i})^2 = \delta_{P_i = Q_i}$; and the third line is true because $\sum_{x_i} \mathrm{Pr}_{\rho, B_{Q_i}}[x_i] = 1$.

Now, the claimed result follows from the Bernstein inequality and a union bound over the at most $(3n)^k$ Pauli operators of degree lower than $k$. $\qquad\square$

**Theorem 3.2.** *Let $\rho$ be a $n$-qubit state whose Pauli spectrum is $(\varepsilon^2/2^{2n})$-concentrated on a set of $k$ qubits. Then, using $O(\frac{12^k \log((3n)^k/\delta)}{\varepsilon^2})$ copies of $\rho$ one can output $\rho'$ such that with success probability $\geq 1 - \delta$ satisfies*

$$\sum_{P \in \{I,X,Y,Z\}^{\otimes n}} |\widehat{\rho'}(P) - \widehat{\rho}(P)|^2 \leq \frac{\varepsilon^2}{2^{2n}}.$$

*In particular, $\|\rho' - \rho\|_{\mathrm{tr}} \leq \varepsilon$. Furthermore, the algorithm just does Pauli measurements on single copies of the state.*

*Proof.* Similarly to the the proof of classical case Theorem 1.1, we use $T = O(\frac{12^k \log((3n)^k/\delta)}{\varepsilon^2})$ copies of the state obtain an estimate $\widehat{\rho'}(P)$ for every $P$ with $|\mathrm{supp}(P)| \leq k$ such that

$$|\widehat{\rho}(P) - \widehat{\rho'}(P)| \leq \frac{\varepsilon}{4\sqrt{4^k}2^n}. \tag{16}$$

This can be done via Classical Shadows (see Theorem 3.1). Now, for every $P \in \{I, X, Y, Z\}^{\otimes n}$ we define

$$\widehat{\rho''}(P) = \begin{cases} 0 & |\mathrm{supp}(P)| > k, \\ 0 & |\widehat{\rho'}(P)| \leq \frac{\varepsilon}{(2 \cdot 2^n \cdot \sqrt{4^k})}, \ |\mathrm{supp}(P)| \leq k, \\ \widehat{\rho'}(P) & \text{otherwise.} \end{cases}$$

In particular, from Equation (16) it follows that for every $S$ with $|\mathrm{supp}(S)| \leq k$ we have that

$$|\widehat{\rho}(P) - \widehat{\rho''}(P)| \leq \frac{\varepsilon}{2^n \sqrt{4^k}}. \tag{17}$$

In addition, we claim that for every $P \in \{I, X, Y, Z\}^{\otimes n}$

$$|\widehat{\rho}(P) - \widehat{\rho''}(P)| \leq |\widehat{\rho}(P)|. \tag{18}$$

Indeed, the only non-trivial case of Equation (18) corresponds to $P$ with $|\mathrm{supp}(P)| \leq k$ and $|\widehat{\rho'}(P)| \geq \varepsilon/(2 \cdot 2^n \cdot \sqrt{4^k})$. In that case, we have that

$$\begin{aligned} |\widehat{\rho}(P)| &\geq |\widehat{\rho'}(P)| - |\widehat{\rho}(P) - \widehat{\rho'}(P)| \\ &\geq \varepsilon/\left(4 \cdot 2^n \cdot \sqrt{4^k}\right) \\ &\geq |\widehat{\rho}(P) - \widehat{\rho'}(P)| \\ &= |\widehat{\rho}(P) - \widehat{\rho''}(P)|, \end{aligned}$$

where the first is due to triangle inequality, the second line is true because of Equation (16) and the hypothesis on $P$, the third line again follows from Equation (16), and the fourth line is true because of the choice of $P$ and the definition of $\widehat{\rho''}(P)$.

Finally, we claim that $\rho'' = \sum_P \widehat{\rho''}(P) P$ is a good approximation to $\rho$. Indeed, let $K \subseteq [n]$ be the subset of qubits where the spectrum of $\rho$ is concentrated on, then

$$\begin{aligned} \sum_{P \in \{I,X,Y,Z\}^{\otimes n}} |\widehat{\rho}(P) - \widehat{\rho''}(P)|^2 &= \sum_{P \in \{I,X,Y,Z\}^{\otimes K}} |\widehat{\rho}(P) - \widehat{\rho''}(P)|^2 \\ &\quad + \sum_{P \notin \{I,X,Y,Z\}^{\otimes K}} |\widehat{\rho}(P) - \widehat{\rho''}(P)|^2 \\ &\leq \sum_{P \in \{I,X,Y,Z\}^{\otimes K}} \frac{\varepsilon^2}{2^{2n}4^k} + \sum_{P \notin \{I,X,Y,Z\}^{\otimes K}} |\widehat{\rho}(P)|^2 \\ &\leq 2\frac{\varepsilon^2}{2^{2n}}, \end{aligned}$$

where in the second line we have used Equations (17) and (18); and in the third line that $|\{I, X, Y, Z\}^{\otimes K}| = 4^k$ and that the spectrum of $\rho$ is $(\varepsilon^2/2^{2n})$-concentrated on $K$. $\qquad\square$

**Theorem C.1.** *To learn $n$-qubit $k$-junta states, one requires $\Omega(4^k/\varepsilon^2)$ copies of the state.*

*Proof.* We will provide a set of $n$ states $\{\rho_i\}_{i\in[n]}$ of $n$ qubits that are 1-junta, and satisfy

$$\|\rho_i - \rho_j\|_{\mathrm{tr}} \geq \varepsilon \text{ if } i \neq j, \tag{19}$$

and

$$\chi\big(\{\rho_i^{\otimes T}\}\big) \leq T\varepsilon^2 \text{ for every } T \in \mathbb{N}. \tag{20}$$

From Equations (19) and (20) the lower bound $\Omega(\log(n)/\varepsilon^2)$ follows from Lemma A.1. For every $\varepsilon \in (0, 1/2)$, we define

$$\rho_\varepsilon = \frac{1}{2}\begin{pmatrix} 1+\varepsilon & 0 \\ 0 & 1-\varepsilon \end{pmatrix}.$$

For $i \in [n]$, we define

$$\rho_i = \frac{I}{2} \otimes \cdots \otimes \underbrace{\rho_\varepsilon}_{i\text{-th qubit}} \otimes \cdots \otimes \frac{I}{2}.$$

Equation (19) holds because if $i \neq j$, then

$$\|\rho_i - \rho_j\|_{\mathrm{tr}} = \left\| \rho_\varepsilon \otimes \frac{I}{2} - \frac{I}{2} \otimes \rho_\varepsilon \right\|_{\mathrm{tr}}$$

$$= \left\| \frac{1}{2}\begin{pmatrix} 0 & & & \\ & \varepsilon & & \\ & & -\varepsilon & \\ & & & 0 \end{pmatrix} \right\|_{\mathrm{tr}}$$

$$= \varepsilon.$$

Proving Equation (20) requires just a bit more work. We begin by noting that

$$|S(\rho_\varepsilon) - 1| \leq O(\varepsilon^2) \tag{21}$$

for every $\varepsilon < 1/2$. Indeed,

$$|S(\rho_\varepsilon) - 1| = \left| -\sum_{x\in\{\pm 1\}} \frac{1+x\varepsilon}{2} \log\left(\frac{1+x\varepsilon}{2}\right) - 1 \right|$$

$$= \left| \sum_{x\in\{\pm 1\}} \frac{1+x\varepsilon}{2} \log(1+x\varepsilon) \right|$$

$$= \left| \sum_{x\in\{\pm 1\}} \frac{1+x\varepsilon}{2}\big(x\varepsilon + O(\varepsilon^2)\big) \right|$$

$$= O(\varepsilon^2),$$

where in the second line we have applied Taylor's theorem. We recall that the Holevo information is given by

$$\chi\big(\{\rho_i^{\otimes T}\}\big) = \underbrace{S\left( \frac{1}{n}\sum_{i\in[n]} \rho_i^{\otimes T} \right)}_{(*)} - \underbrace{\frac{1}{n}\sum_{i\in[n]} S\big(\rho_i^{\otimes T}\big)}_{(**)}.$$

We will analyze the terms $(*)$ and $(**)$ separately. We begin with $(**)$:

$$(**) = S\big(\rho_1^{\otimes T}\big) = T(S(\rho_\varepsilon) + (n-1)) \geq Tn - O\big(T\varepsilon^2\big),$$

where we have applied additivity of the entropy under the tensor product (see Equation (13)) and Equation (21). The analysis of the term $(*)$ is a bit more involved:

$$
\begin{aligned}
(*) &= S\left(\frac{1}{n}\left\{\rho_\varepsilon^{\otimes T} \otimes \left(\frac{I}{2}\right)^{\otimes T} \otimes \cdots \otimes \left(\frac{I}{2}\right)^{\otimes T} + \cdots + \left(\frac{I}{2}\right)^{\otimes T} \otimes \cdots \otimes \rho_\varepsilon^{\otimes T}\right\}\right) \\
&\leq nS\left(\frac{1}{n}\left\{\rho_\varepsilon^{\otimes T} + (n-1)\left(\frac{I}{2}\right)^{\otimes T}\right\}\right) \\
&\leq nTS\left(\frac{1}{n}\left\{\rho_\varepsilon + (n-1)\frac{I}{2}\right\}\right) \\
&= nTS(\rho_{\frac{\varepsilon}{n}}) \\
&\leq nT + TO(\varepsilon^2/n^2),
\end{aligned}
$$

where in the second and third lines we have applied subadditivity of the entropy (see Equation (14)), and in the last line Equation (21). Putting the analysis for terms $(*)$ and $(**)$ together, Equation (20) follows. $\square$

