# OpenReview forum: "Learning Junta Distributions, Quantum Junta States, and QAC$^0$ Circuits"
_ICML.cc/2026/Conference — ICML 2026 regular_

### Official Review · Reviewer_NMtH · 2026-03-11

**Soundness:** 3
**Presentation:** 3
**Significance:** 3
**Originality:** 3
**Overall Recommendation:** 4
**Confidence:** 3

**Summary:**

Juntas are natural models in learning theory for objects with concise structure that can be leveraged by learning algorithms. This work studies learning and testing of juntas, a novel quantum state analog of them, and QAC^0 circuits, and provides novel sample complexity results, improving on previous works. In some cases the results are optimal.

**Compliance With Llm Reviewing Policy:**

Affirmed.

**Final Justification:**

I would like to retain my score.

**Key Questions For Authors:**

The authors introduce a novel notion of junta states that generalize classical juntas. Are there physical systems that could conceivably produce such states, or do the authors believe they are mostly of theoretical interest?

**Limitations:**

yes

**Strengths And Weaknesses:**

This is a nice contribution to learning theory. Some of the sample complexity results match known lower bounds exactly.
Intermediate results such as the improved analysis of classical shadows may be of independent interest, since this is a widely used technique.

The presentation is clear and concise. There is a broken reference on page 2.

The work appears to use standard techniques, though the optimality of the results gives them increased significance.

---

> ### Author Rebuttal · Authors · 2026-03-30
>
> Dear reviewer,
>
> We would like to thank you for your time spent reviewing our manuscript and for the thoughtful comments on our paper.
> We will fix the broken reference. Thanks for pointing it out.
>
> Regarding examples close to junta states, the Choi states of QAC$^0$ circuits are one of them, but not the only. For instance, states prepared by shallow circuits are relevant states that are also close to junta states. Furthermore, current quantum devices typically have only a few clean qubits, and the rest are very noisy. Hence, the states of these systems are naturally close to junta states, where the relevant qubits are the clean qubits, and the noisy qubits are effectively maximally mixed.
>
> These examples show that our results for quantum junta states can provide insights for learning in realistic quantum systems.

---

> > ### Author Rebuttal · Reviewer_NMtH · 2026-04-02
> >
> > I thank the authors for their rebuttal. I would like to keep my score.

---

### Official Review · Reviewer_Xgy5 · 2026-03-11

**Soundness:** 3
**Presentation:** 3
**Significance:** 4
**Originality:** 4
**Overall Recommendation:** 5
**Confidence:** 4

**Summary:**

The paper studies learning problems for three related structured objects: classical junta distributions, a newly introduced notion of quantum junta states, and QAC^0 circuits. Its main contribution is to show that these objects can be learned efficiently by exploiting the fact that their Fourier or Pauli representations are both low-degree and sparse, yielding an essentially optimal sample bound for learning classical junta distributions and near-matching upper and lower bounds for learning quantum junta states. For QAC^0, the paper argues that the Choi states of such circuits are not just low-degree but also close to juntas, which leads to improved copy complexity for learning them from Choi states. Along the way, it also gives a new Pauli-analysis-based proof of the performance of classical shadows and strengthens lower bounds for QAC^0 on the address function.

**Compliance With Llm Reviewing Policy:**

Affirmed.

**Key Questions For Authors:**

1) Could you clarify the exact operational meaning of the QAC^0 learning guarantee for Choi states, especially how the normalized Frobenius-error guarantee should be interpreted relative to more standard channel-distance notions?
A clearer discussion here would help one judge how broadly meaningful the QAC^0 result is beyond the formal learning model studied in the paper. If you can justify that this metric is the right one for the intended applications, or explain whether comparable results are plausible in stronger metrics, that would strengthen assessment of the significance.

2) In the QAC^0 section, can you give a more explicit high-level explanation of why “close to junta” is strictly stronger than prior low-degree concentration results, and which step in the learning improvement fundamentally uses this stronger structure?
I believe this is one of the paper’s most original ideas, but the distinction is somewhat compressed in the current presentation. A convincing explanation would increase confidence in both the originality and the significance of this part of the paper.

**Limitations:**

No. The paper includes a brief impact statement, but it is quite minimal and does not meaningfully discuss limitations of the results. A stronger discussion would note the main scope limitations explicitly. For example, the highly theoretical setting, the specialized learning models and error metrics, and the fact that the practical implications for near-term ML applications are limited.

**Strengths And Weaknesses:**

The paper appears technically sound, with the main upper and lower bounds fitting standard and appropriate techniques from learning theory, Fourier/Pauli analysis, and quantum information. The QAC^0 results are the most intricate, but the argument that Choi states are close to juntas seems coherent and well aligned with the claimed improvements. The main weakness is that some of the more delicate proof ideas are compressed, which makes the hardest steps somewhat difficult to verify quickly.

It is well structured overall, and the unifying theme across junta distributions, quantum junta states, and QAC^0 gives it a coherent narrative. However, the exposition is quite dense, and some of the main intuitions, especially for quantum junta states and the QAC^0 closeness result, could be explained more clearly in the main text. A bit more proof roadmap and notation guidance would make the paper easier to follow.

This is a meaningful contribution for theoretical machine learning and quantum learning theory: it gives essentially optimal or near-optimal learning results in several settings and introduces a natural new object in quantum junta states. The QAC^0 structural result also seems likely to be useful beyond the immediate theorem, since it strengthens the understanding of what these circuits look like spectrally. The impact is specialized rather than broad, but within its area it is substantial.

The most original aspects are the introduction of quantum junta states and the observation that QAC^0 Choi states are not just low-degree but close to juntas, which leads to stronger learning consequences. The Pauli-analysis perspective also gives a fresh conceptual angle, including a new proof related to classical shadows. The classical junta-distribution result feels more like a sharp refinement of existing ideas than a wholly new technique, but it is still a valuable and nontrivial contribution.

---

> ### Author Rebuttal · Authors · 2026-03-30
>
> Dear reviewer,
>
> We would like to thank you for your time spent reviewing our manuscript and for the thoughtful comments on our paper.
>
>
> We start by motivating the choice of the normalized Frobenius norm. The use of this metric is a natural measure of closeness when using QAC$^0$ circuits to compute Boolean functions. Concretely, consider that one had a quantum circuit that computed a given Boolean function $f$: {0,1}$^n \to$ {0,1}. Consider also a QAC$^0$ circuit computing $g$ : {0,1}$^n \to$ [0,1], where $g(x)$ is the probability of measuring 1 on input $x$. Then, the probability that $f$ differs from the rounding of $g$ to {0,1}, over a uniformly random input, is upper bounded by the normalized Frobenius distance of the circuits. Intuitively, this means that learning the Choi state in the normalized Frobenius norm implies learning the classical computation. While other metrics for quantum channels, such as the diamond norm, are more relevant in other contexts, in this setting, the normalized Frobenius norm is the best suited.
>
>
> Next, we explain why being close to the junta implies being close to low-degree, and not the other way around. By definition, $k$-junta objects have degree at most $k$, but the converse is not true. An example of that is the memory address function, which has degree $d$, but is not close to any $d$-junta, as we show in our manuscript. Furthermore, the memory address function can be embedded into a quantum state (by setting the diagonal to the truth table, and normalizing), which illustrates why degree $d$ does not imply the $d$-junta property also in the quantum setting.
>
> In particular, $k$-juntas are $2^k$-sparse objects, which is crucial in our learning algorithms. However, this is not true for degree $d$ distributions or degree $d$ quantum states, which can have sparsity of the order $n^d$. Thus, the close-to-junta feature allows us to improve over prior work on learning QAC$^0$ circuits.
>
> In conclusion, the normalized Frobenius norm captures the relevant operational behavior of QAC$^0$ circuits when performing classical computations, and the key enabler of our improvement over prior work on QAC$^0$ circuits is the $k$-junta property.

---

> > ### Author Rebuttal · Reviewer_Xgy5 · 2026-04-01
> >
> > My concerns have been adequately addressed.

---

### Official Review · Reviewer_jE8y · 2026-03-12

**Soundness:** 4
**Presentation:** 3
**Significance:** 2
**Originality:** 3
**Overall Recommendation:** 4
**Confidence:** 2

**Summary:**

This paper investigates the problems of learning classical junta distributions, quantum junta states, and $QAC^0$ circuits. For classical junta distributions, the authors present an algorithm that requires $O(2^k \log(n)/\epsilon^2)$ samples. This quadratically improves upon previous upper bounds and matches existing lower bounds in every parameter. Additionally, the paper initiates the study of $n$-qubit quantum junta states, which are the tensor product of a $k$-qubit state and an $(n-k)$-qubit maximally mixed state. The authors provide upper and lower bounds for learning and testing these quantum states. Finally, the authors show that the Choi states of $QAC^0$ circuits are close to being juntas, which they leverage to provide improved learning upper bounds for these circuits.

**Compliance With Llm Reviewing Policy:**

Affirmed.

**Key Questions For Authors:**

Can you give a short explanation on the relevance to machine learning?

**Limitations:**

yes

**Strengths And Weaknesses:**

Strengths:
The paper is well written and organized.The mathematical derivations and proofs that I have checked are sound.The upper bounds provided rely on the clever observation that the Fourier and Pauli expansions of these objects are both low-degree and sparse.

Weaknesses & Limitations:
The main limitation IMO is that it is hard for me to accurately assess the contribution to learning theory. Lacking  background background in Quantum Computations, I cannot confidently evaluate the broader impact of the quantum junta states and $QAC^0$ findings.

Because it is hard for me to access the contribution to the learning theory of quantum-related distributions, I must keep my confidence low. I vote for a Weak Accept, but I will defer to other reviewers with deeper expertise in quantum learning theory to validate the significance of these specific results.

---

> ### Author Rebuttal · Authors · 2026-03-30
>
> Dear reviewer,
>
> We would like to thank you for your time spent reviewing our manuscript and for the thoughtful comments on our paper.
>
> Our work is relevant for machine learning, as it rigorously addresses the problem of learning high-dimensional data in the presence of structure.
>
> In the classical case, we study the problem of learning high-dimensional junta distributions. This is closely related to the sparsity assumptions and feature selection problem prevalent in machine learning.
>
> In the quantum case, with the rapid recent development of quantum computers, the problem of characterizing the behavior of quantum devices, i.e., learning quantum objects, has become pressing. Here, quantum junta states capture an important example of high-dimensional states with few degrees of freedom. They approximate several families of practically relevant states, such as those prepared by shallow circuits and low-entanglement states. Our quantum findings fit into the newly emerging field of quantum machine learning.
>
> To conclude, we stress that our results provide improved, sometimes optimal, guarantees for these problems, belonging to the theoretical branch of machine learning.

---

> > ### Author Rebuttal · Reviewer_jE8y · 2026-04-01
> >
> > Thanks for the authors. I will keep my score

---

### Official Review · Reviewer_Z3Tk · 2026-03-12

**Soundness:** 2
**Presentation:** 3
**Significance:** 2
**Originality:** 3
**Overall Recommendation:** 4
**Confidence:** 3

**Summary:**

The paper studies the efficient learning of structured objects in computational and quantum learning theory, including classical junta distributions, quantum junta states, and QAC0 circuits.

The authors first improve the sample complexity for learning classical k-junta distributions and achieve a quadratic improvement over prior work. They then introduce the notion of quantum junta states, where only k qubits contain information while the remaining qubits are maximally mixed, and provide nearly optimal bounds for learning and testing such states using Pauli measurements and classical shadow techniques. Finally, they show that the Choi states of QAC0 circuits are close to junta states, which enables improved learning algorithms for these circuits.

A key theme of the work is the connection between low-degree and sparse Fourier or Pauli spectra and learnability. By exploiting these structural properties, the paper develops improved learning algorithms and provides new insights into the structure and learnability of quantum circuits and distributions.

**Compliance With Llm Reviewing Policy:**

Affirmed.

**Key Questions For Authors:**

1.The paper focuses mainly on theoretical guarantees. Could the authors comment on whether the proposed algorithms could be evaluated empirically, for example through simulations on small-scale quantum systems or synthetic datasets?
2.The analysis relies on structural assumptions such as sparsity and low-degree Fourier or Pauli spectra. How common are these assumptions in practical learning tasks involving quantum states or circuits?
3.It may help to include additional intuition or illustrative examples to improve accessibility for readers who are less familiar with the area.

**Limitations:**

yes

**Strengths And Weaknesses:**

Strengths
1.The paper provides improved sample complexity bounds for learning classical k-junta distributions and extends the analysis to quantum junta states with nearly optimal guarantees.
2.The work connects several related learning problems, including classical junta distributions, quantum states, and QAC0 circuits, and highlights structural similarities between them.
3.The paper emphasizes the role of sparse and low-degree Fourier or Pauli spectra in learnability, which provides useful theoretical insights.

Weaknesses
1.The work is primarily theoretical and does not include experimental results to illustrate the practical performance of the proposed methods.
2.The results rely on strong structural assumptions such as sparsity and low-degree spectra, which may limit applicability in more general settings.
3. Some parts of the technical exposition are dense and may be difficult to follow without substantial background in quantum learning theory.

---

> ### Author Rebuttal · Authors · 2026-03-30
>
> Dear reviewer,
>
> We would like to thank you for your time spent reviewing our manuscript and for the thoughtful comments on our paper.
>
> Regarding Question 1, indeed, our algorithms could be empirically evaluated. In the classical case, one can generate junta-type datasets and then verify our improvements in the sample complexity. In the quantum case, we could use the available small-scale quantum systems. We acknowledge that running such experiments would be interesting, as they may show that for practical instances, our algorithms may be even more efficient than for the worst-case scenario, for which we prove nearly optimal scalings.
>
> Regarding Question 2, the assumptions of being low-degree and sparse appear in several settings. In the classical case, they fit into the celebrated paradigm of learning in the presence of irrelevant features, initiated by Avrim Blum in the 90's. In the quantum Choi states case, states prepared by shallow circuits or the states of noisy devices with just a few clean qubits are approximately low-degree and sparse. Thus, these assumptions capture a broad and practically relevant regime where efficient learning is possible.
>
> Regarding Question 3, we acknowledge that our work can benefit from a few examples. In the revision, we will include concrete and simple classical and quantum examples illustrating junta structure and sparse Fourier/Pauli spectra, such as those mentioned in the previous paragraph.
>
> Finally, we believe that our work provides a unified framework to study distinct types of junta objects and shallow circuits, which leads us to improve the learning complexity of these objects with respect to several prior works.

---

> > ### Author Rebuttal · Reviewer_Z3Tk · 2026-04-03
> >
> > Thanks for the authors. I will keep my score

---

### Decision · Program_Chairs · 2026-04-30

**Decision:**

Accept (regular)

**Comment:**

This paper studies the learning of classical junta distributions, quantum junta states (a new notion introduced here), and QAC^0 circuits. It achieves optimal or near-optimal sample complexity bounds across all three settings by exploiting the joint low-degree and sparse structure of Fourier/Pauli representations.

All four reviewers support acceptance (scores: 4, 4, 4, 5) and all concerns were resolved in the rebuttal. Reviewer Xgy5 (confidence 4) highlights the introduction of quantum junta states with likely utility beyond the immediate theorems. The paper is well-written and provides a coherent unifying narrative across the three settings.